# Identification of antimicrobial peptides from the *Ambystoma mexicanum* displaying antibacterial and antitumor activity

**Nadjib Dastagir**[1]*, **Christina Liebsch**[1], **Jaqueline Kutz**[1], **Sabine Wronski**[2],
**Andreas Pich**[3], **Doha Obed**[1], **Peter Maria Vogt**[1], **Vesna Bucan**[1●], **Sarah Strauß**[1●]

1 Department of Plastic, Aesthetic, Hand and Reconstructive Surgery, Hannover Medical School, Hannover, Germany 2 Fraunhofer Institute for Toxicology and Experimental Medicine, Hannover, Germany, 3 Hannover Medical School, Institute for Toxicology, Hannover, Germany

● These authors contributed equally.
* nadjibdastagir@gmail.com

## Abstract

Antibiotic resistance is a significant healthcare concern. Therefore, identifying target molecules that can serve as antibiotic substitutes is crucial. Among the promising candidates are antimicrobial peptides (AMPs). AMPs are defense mechanisms of the innate immune system which exist in almost all living organisms. Research on the AMPs of some amphibians has shown that, in addition to their antimicrobial effectiveness, AMPs also exhibit anti-inflammatory and anti-carcinogenic properties. In this study, we identify and characterize AMPs deriving from the skin mucus of the axolotl (*Ambystoma mexicanum*). Upon activity spectrum evaluation of the AMPs, we synthesized and ranked 22 AMPs according to antimicrobial efficacy by means of a prediction tool. To assess the AMPs' potential as antibacterial and anticarcinogenic compounds, we performed a minimum inhibitory concentration (MIC) assay for efficacy against methicillin-resistant *Staphylococcus aureus* (MRSA) and methicillin-sensitive *Staphylococcus aureus* (MSSA), and an apoptosis assay on T-47D mammary carcinoma cells. We identified four AMPs that showed significant inhibition of MRSA, of which three also demonstrated anticarcinogenic activity. Gene expression analysis was performed on AMP-stimulated carcinoma cells using a breast cancer-specific RT-PCR array. In cells stimulated with the AMPs, gene expression analysis showed upregulation of tumor suppressor genes and downregulation of oncogenes. Overall, our work demonstrates the antimicrobial and anticarcinogenic activity of axolotl-derived AMPs. The results of this work serve as a basis to further investigate the mode of action and potential use of axolotl AMPs as therapeutic anticancer or antibiotic agents.

## Introduction

Antibiotic resistance is a global public health problem associated with increased mortality and morbidity [1]. The ability of bacteria to rapidly evolve and become resistant to conventional antibiotics creates an urgent need to identify new targets that can be used as substitutes [2]. One possible solution could be the use of antimicrobial peptides (AMPs), also known as host

**Data availability statement:** The datasets used and/or analyzed during the current study are, within the manuscript itself, and uploaded as supplementary.

**Funding:** The author(s) received no specific funding for this work.

**Competing interests:** The authors have declared that no competing interests exist.

defense peptides (HDPs). AMPs are part of the innate immune response and are found in many species across the phylogenetic kingdoms [3]. These peptides function as "endogenous antibiotics" with particular importance in the early defense against bacteria, as well as viruses, fungi, and parasites [4–6]. AMPs are either constitutively expressed or produced in response to infection and inflammatory stimuli. They consist of sequences less than 100 amino acids long with a net positive charge [7].

Similar to the complement system, AMPs act primarily through mechanisms involving membrane disruption, reducing their risk of developing multidrug resistance, and are capable of inhibiting some antibiotic-resistant microorganisms [8,9]. Although AMPs share common properties, such as those mentioned above, they differ in sequence and activity [10]. In addition to their crucial role in inflammatory reactions, immune system activation, and wound healing, they also have an anticarcinogenic effect [11,12]. It is noteworthy that numerous studies have demonstrated the significant selective anticarcinogenic activity of many AMPs, which can also be referred to as anti-cancer peptides (ACPs) [13].

Because AMPs are naturally occurring across many species, many groups have investigated how AMPs isolated from amphibians and arthropods, considered to be organisms with very strong innate immune systems, respond to bacteria [3,14]. In Anurans (frogs) such as *Bombina variegata*, *Phyllomedusa sauvagii* and *Xenopus laevis*, various skin peptides have already been described and tested for their antimicrobial activity [15]. Studies investigating the role of AMPs isolated from *Xenopus laevis* frogs found synergistic modes of action against bacterial strains by forming transmembrane pores, thus resulting in a more powerful inhibition of the bacteria [7,16,17].

Studying the characterization of antimicrobial peptides (AMPs) from amphibians and arthropods can potentially lead to the identification of new target molecules for developing therapies against antibiotic resistance. It is noteworthy that the characterization of AMPs from the axolotl (*Ambystoma mexicanum*) has not yet been explored. This presents a promising opportunity for further research in this area. The innate immune system of axolotls is particularly effective, while their adaptive immune response is weak [18]. The animals produce only two classes of immunoglobulins: IgM and IgY [19]. Thus, the mucus and the nonspecific immune system of axolotls are their main defense mechanism against pathogens. In this regard, antimicrobial peptides in the mucus, neutrophils, and macrophages are mainly responsible for the immune defense [20]. Based on this dependence on the innate immune system, we hypothesized that the AMPs produced by axolotls are particularly effective antimicrobial agents. Amphibian antimicrobial peptides (AMPs) primarily adopt a linear structure and fold into an amphipathic helix that binds to the membrane [21]. These peptides are typically sourced from the skin glands of the animals and are secreted during stress or injury [22,23].

Numerous studies have demonstrated that several antimicrobial peptides (AMPs) exhibit exceptional antitumor activity in vitro and in vivo, particularly against breast cancer and lung cancer [24,25]. AMPs are natural agents that hold great promise as a novel class of anticancer drugs [26,27]. Conventional chemotherapeutic agents have weaknesses, such as lack of selectivity for tumor cells, which can easily lead to severe side effects, and induce tumor cell resistance [28]. AMPs, on the other hand, offer considerable advantages in the fight against tumors, such as a broad spectrum of antimicrobial and antitumor effects, high selectivity for cancer cells, safety for normal cells and vital organs and low resistance formation [29,30].

The selective membrane-destroying effects are similar to the bactericidal effects: The cationic AMPs directly interfere with the membrane of cancer cells by electrostatic attraction to form temporary pores and damage the integrity of the cell membrane, ultimately leading to cell death [31]. The non-membranolytic effects of AMPs include destruction of the cytoskeleton of cancer cells, inhibition of DNA and protein synthesis, inhibition of tumor angiogenesis, immune regulation and induction of apoptosis or tumor cell necrosis [32,33].

Studies indicate that amphibians may have a higher resistance to cancer, despite their high cell proliferation [34,35]. Reasons for this have not yet been described in the literature, however, AMPs might be attributed to this observation. Certain amphibian AMPs were found to have anticarcinogenic activities. Magainins, aureins, cytropins, and peptides of brevinin-1, ranatuerin-2, temporin, and peptides of the dermaseptin family show anticarcinogenic activity at doses that exhibit little or no toxicity against regular mammalian cells [23].

Among the best studied "antitumor peptides" are the ionophore magainins from *X. laevis* [36]. Magainin 2 and its synthetic analogs kill a wide range of cancer cell lines including lung, breast, and bladder cancer cells, as well as cells from lymphomas, melanomas, and leukemias [23]. The anticarcinogenic effect of Magainin 2 has also been demonstrated in vivo using subcutaneous xenografts in nude mouse models [23], suggesting AMPs isolated from amphibians have the potential to recapitulate their activities in mammalian models.

In this work, we characterize AMPs deriving from the skin mucus of the axolotl and use synthesized analogs to assess the potential of these AMPs as antibacterial and anti-cancer therapies. The aim of our study is to demonstrate the use of axolotl-derived AMPs in inhibiting MRSA, as well as their translational anticarcinogenic activity in mammalian tumor cell lines. This study identified antimicrobial peptides (AMPs) that have potential as both anticancer agents and for future research on antibiotic resistance.

## Materials and methods

### Animal husbandry and mucus harvesting

The axolotls used in this study were originally obtained from the Ambystoma Genetic Stock Center (AGSC) at the University of Kentucky. The AGSC maintains a colony of axolotls and provides them to research laboratories worldwide, along with housing, breeding and care information. All experiments were performed in accordance with the guidelines of the German Animal Welfare Act. Due to these guidelines and after consultation of the veterinary inspection office as well as the "Landesamt für Lebensmittelsicherheit und Verbraucherschutz" (LAVES, relevant authority for animal trials in lower Saxony, Germany) mucus harvesting from the animals was classified as non invasive and therefore not requiring a permission. All methods are reported in accordance with ARRIVE guidelines. The axolotls were kept in tap water and fed a specialized diet (Axobalance; AquaTerratec, Bröckel, Germany) twice a week. The animals were housed in groups of 3 to 5 in fully equipped tanks (ground substrate, hiding places, artificial plants, filter system) without artificial illumination. Circadian rhythm was given by common daylight without direct exposition to sunlight through windows. Keeping temperature ranged from 12°C in winter to max. 20°C in summer. To obtain the skin mucus containing the AMPs to be examined, axolotls were gently massaged with sterile nitrile gloves, and the produced mucus was collected with sterile scrapers (Fig 1).

### Mass spectrometric analysis and peptide synthesis

The chromatography and mass spectrometric analysis were conducted using an AXIMA Performance MALDI-TOF/TOF mass spectrometer (Shimadzu) at the Institute for Toxicology in Hannover. Skin secretion samples were processed by mass spectrometry and analyzed by the Proteomics Facility of Hannover Medical School. Fractions were dissolved in Tris-buffer or in 30% acetonitrile and 0.1% TFA. The contained peptides were subsequently loaded onto a C-18 analytical column (Vydac 238 TP, 150 × 4.6 mm) and solubilized by slowly increasing the acetonitrile concentration up to 70% over a time of 120 minutes at 1 mL/min flow rate. The absorbance of the eluted solution was measured at 214nm and 280nm. The obtained fractions were analyzed by liquid chromatography-mass spectrometry. For the chromatography,

**Fig 1. Schematic for the isolation procedure of AMPs from axolotl skin secretions.** Axolotls were gently massaged with nitrile gloves to collect skin mucus. AMPs were then isolated from the mucus by liquid chromatography-mass spectrometry. The identified peptide sequences were ranked according to their predicted antimicrobial activity using bioinformatics tools, and the 22 most promising sequences were synthesized.

standard injection volumes of 5 µL were used. The obtained fractions were subsequently analyzed by liquid chromatography-mass spectrometry. Peptide identification was carried out using the instrument's integrated software, which supports fully automated proteomics experiments and LC-MALDI analyses.

We used the CAMP$_{R3}$ (Collection of Anti-Microbial Peptides) database and its classifiers (SVM, RF, ANN, and DA) to predict the antimicrobial potential of the identified peptides. We focused on peptides that showed high probability of antimicrobial activity across multiple classifiers. The final 22 peptides were selected based on their consistently high predicted antimicrobial activity and other promising bioactive properties. The 22 most promising sequences were synthesized by the commercial supplier Caslo Laboratory (Denmark). The purity of the peptides was also determined by Caslo using high performance liquid chromatography (HPLC). The peptides were named 1 to 22 according to their ranking in the prediction tool. 1 represents the peptide with the highest predicted efficacy, 22 the one with the lowest included in the experiment. The synthetically produced peptides were lyophilized and reconstituted in DMSO (Merck SA, Germany) acetic acid (Roth, Germany), water (Merck Millipore, Germany) or Tris-buffer (Merck SA, Germany) depending on their solubility.

## Screening of antimicrobial activity

MIC assays (MRSA, ATCC# 43300/ MSSA, ATCC# 6538) were performed according to CLSI guidelines. (Committee for Clinical Laboratory Standards Methods for Dilution Antimicrobial Susceptibility Tests for Bacteria That Grow Aerobically—Tenth Edition: Approved Standard M7-A10. NCCLS, Wayne, PA, USA, 2015.) In short, 96-well plates were inoculated with 100µL of bacterial suspension and 100µL of individual peptides or appropriate diluent (DMSO, acetic acid, water or Tris-buffer depending on peptide solubility) as a negative control, and samples were cultured under static conditions with 100% humidity, 37°C and 0% $CO_2$ for 24 h. Thereafter, samples were treated with peptide concentrations ranging from 10, 5, 2.5, 2, 1.75, 1.5 1.25, 1, 0,85, 0.625 and 0.313 µg/mL. Human antimicrobial peptide LL-37 was utilized as a positive control at a concentration of 256 µg/mL, as it is one of the best-known examples of AMP-based drug development. The assay was performed in triplicates. Efficacy was assessed according to the viable bacterial load determined by dilution plating and counting of Colony Forming Units (CFU). The lowest concentration that inhibited bacterial growth after 24 h of incubation is the MIC. Growth curve were performed using a plate reader (Synergy 2, Biotek).

## Cell culture

The human breast cancer cell line T-47D (HTB-133) was purchased from ATCC. Cells were maintained in RPMI-1640 medium (Thermo Fisher Scientific catalog number 11875093) supplemented with 0.2 units/mL insulin and 10% fetal calf serum (FCS; catalog number

16000044,). The control cell line MCF10A (ATCC, Rockville, MD, USA) was cultured in DMEM/F12 medium (Gibco) supplemented with 5% horse serum (Sigma), 100 ng/ml cholera toxin (Sigma), 20 ng/ml epidermal growth factor (Sigma), 0.01 mg/ml insulin (Sigma) and 500 ng/ml hydrocortisone (Sigma). Cells were incubated at 37°C in a humidified atmosphere containing 5% $CO_2$. Possible apoptotic effects as a result of the diluent were accounted for using a negative control of the appropriate diluent (DMSO, acetic acid, water, or Tris-buffer).

## Apoptosis assay

The apoptosis assay was performed using the Apo-ONE® Homogeneous Caspase-3/7 Assay Kit according to the manufacturer's instructions (Promega). Briefly, T47D and MCF10A cells were seeded on 96-well plates. Cells were counted using the CountessTM II Automated Cell Counter (Invitrogen). In total, $1 \times 10^6$ cells were treated. After 24 hours, the medium was replaced with fresh medium, and the 22 peptides were added to individual wells at 1, 2.5, 5, and 10 μg/ml. One plate was left without peptide stimulation as a control. After 24 hours of incubation, 79 μL of Caspase Substrate Z-DEVD-R110 (100X) was added to 7.9 mL of Apo-ONE® Homogeneous Caspase-3/7 Buffer, 100 μL of this reagent was added per well. After 60 minutes, fluorescence was measured using the Tecan microplate reader excitation wavelength range of 485 ± 20nm and an emission wavelength range of 530 ± 25nm. (Tecan Group, Schweiz). A Student's t-test was used to compare the two groups.

## Antitumor activity gene expression assay

Three 75 cm² cell culture flasks containing the T-47D and three 75 cm² cell culture flasks containing the MCF10A cells were individually stimulated with peptides 1, 12, or 13 at a concentration of 10 μg/mL. An unstimulated cell culture flask was left as a control. Cells were incubated for 24 hours according to standard protocol at 37°C and 5% $CO_2$, in a humidified atmosphere.

After 24 hours, the cells were detached, and RNA was isolated according to the Macherey-Nagel kit. RNA quality was assessed using a NanoDrop 1000 spectrophotometer and gel electrophoresis. To create cDNA the iScript cDNA Synthesis Kit (BioRad) was used and followed according to manufacturer's instructions. For gene expression analysis, the Human Breast Cancer RT2 Profiler PCR array was used (Qiagen) according to manufacturer's instructions. On this plate, forward and reverse primers were already included in each well for the respective genes. The qPCR cycling conditions were as follows: 8.5 min at 95°C, 40 min at 55°C, 30 min at 72°C and 4°C as holding temperature. Housekeeping gene stability was validated, and results were geometrically averaged. Beta (β)-actin was used for endogenous controls. Expression of carcinogenic markers was quantified relative by the ΔΔCt method and results were normalized to housekeeping genes (Livak and Schmittgen, 2001). The Ct values of the untreated samples were set to zero. The Ct values of the MCF10A samples were subtracted from the Ct values of the T-47D samples. Relative gene expression was calculated from the zero line using Microsoft Excel Version 2016 software (Microsoft Cooperation, Redmond, WA, USA). The resulting higher or lower expressions of the analyzed genes were presented. Gene expression was measured using the BioRad i-Cycler.

## Statistics

Statistical comparisons between groups were performed using an independent t-test, with all analyses conducted using GraphPad Prism 9.2.0. Differences were considered statistically significant at $p < 0.05$, with significance levels denoted as follows: * $p \leq 0.05$, ** $p \leq 0.01$, *** $p \leq 0.001$, **** $p \leq 0.0001$.

## Results

### Mass spectrometric analysis

Mass spectrometric analysis of the fractionated axolotl skin mucus revealed 4,986 peptide sequences. These sequences were analyzed for the probability of antimicrobial activity using the $CAMP_{R3}$ Collection of Anti-Microbial Peptides prediction tool. We selected the 22 most promising peptide candidates based on the calculated probabilities (Table 1, ranked in order of highest to lowest predicted efficacy). BLASTp analysis revealed no significant sequence homology for peptides 1 and 12 (E-value > 0.01). Peptide 13 showed weak alignments with AMPDB_42975|A0A2V2ABU9 (E-value = 3.7) and AMPD-B_41234|A0A2A2DWC7 (E-value = 7.1), indicating statistically insignificant similarities likely occurring by chance.

### Screening for antimicrobial activity

We evaluated the antimicrobial activity of the synthesized peptides by assessing their MIC against Meticillin-Sensitive Staphylococcus aureus and Methicillin-resistant Staphylococcus aureus (MSSA and MRSA). The results of the growth inhibition test showed that peptide 1 had the greatest inhibitory effect on both MSSA and MRSA (Fig 2A). Inhibitory effects against MSSA were also seen by peptide 1, 2, 13, 7, 8 and 3, in order of lowest to highest MIC. For MRSA, an inhibitory effect was seen by peptides 1, 2, 13 and 7, in order of lowest to highest MIC. The MIC of vancomycin of 2 μg/mL was used as a threshold for identifying inhibitory effects.

**Table 1. Amino acid composition and purity of peptide lots P130819-01-01 to P130819-01-22.**

| Peptid Lot Nr.: | Quantity in mg | Hydrophobic amino acids % | Acidic amino acids % | Alkaline amino acids % | Neutral amino acids % | Purity |
|---|---|---|---|---|---|---|
| P130819-01-01 | 5.5 | 57.89 | 0.00 | 5.26 | 36.84 | 95.16% |
| P130819-01-02 | 5:0 | 78.57 | 14.29 | 7.14 | 0.00 | 98.50% |
| P130819-01-03 | 5.5 | 47.62 | 14.29 | 4.76 | 33.33 | 98.72% |
| P130819-01-04 | 5.5 | 63.16 | 0.00 | 5.26 | 31.58 | 98.62% |
| P130819-01-05 | 5.5 | 58.62 | 13.79 | 3.45 | 24.14 | 96.25% |
| P130819-01-06 | 5.5 | 60.87 | 0.00 | 4.35 | 34.78 | 97.20% |
| P130819-01-07 | 5.5 | 41.67 | 0.00 | 8.33 | 50.00 | 96.70% |
| P130819-01-08 | 5.5 | 71.43 | 0.00 | 14.29 | 14.29 | 95.57% |
| P130819-01-09 | 5.0 | 50.00 | 5.00 | 5.00 | 40.00 | 95.35% |
| P130819-01-10 | 5.0 | 56.52 | 8.70 | 4.35 | 30.43 | 98.12% |
| P130819-01-11 | 5.0 | 45.00 | 5.00 | 5.00 | 45.00 | 97.91% |
| P130819-01-12 | 5.0 | 45.45 | 0.00 | 9.09 | 45.45 | 97.00% |
| P130819-01-13 | 5.5 | 57.14 | 4.76 | 9.52 | 28.57 | 95.35% |
| P130819-01-14 | 5.0 | 52.94 | 0.00 | 5.88 | 41.18 | 97.88% |
| P130819-01-15 | 5.0 | 52.63 | 0.00 | 5.26 | 42.11 | 97.35% |
| P130819-01-16 | 5.5 | 50.00 | 0.00 | 10.00 | 40.00 | 99.36% |
| P130819-01-17 | 5.5 | 52.63 | 10.53 | 10.53 | 26.32 | 98.22% |
| P130819-01-18 | 5.0 | 50.00 | 0.00 | 0.00 | 50.00 | 96.95% |
| P130819-01-19 | 5.0 | 50.00 | 0.00 | 6.25 | 43.75 | 97.00% |
| P130819-01-20 | 5.5 | 50.00 | 0.00 | 33.33 | 16.67 | 99.39% |
| P130819-01-21 | 5.5 | 28.75 | 0.00 | 57.14 | 14.29 | 96.96% |
| P130819-01-22 | 5.0 | 16.67 | 0.00 | 50.00 | 33.33 | 96.79% |

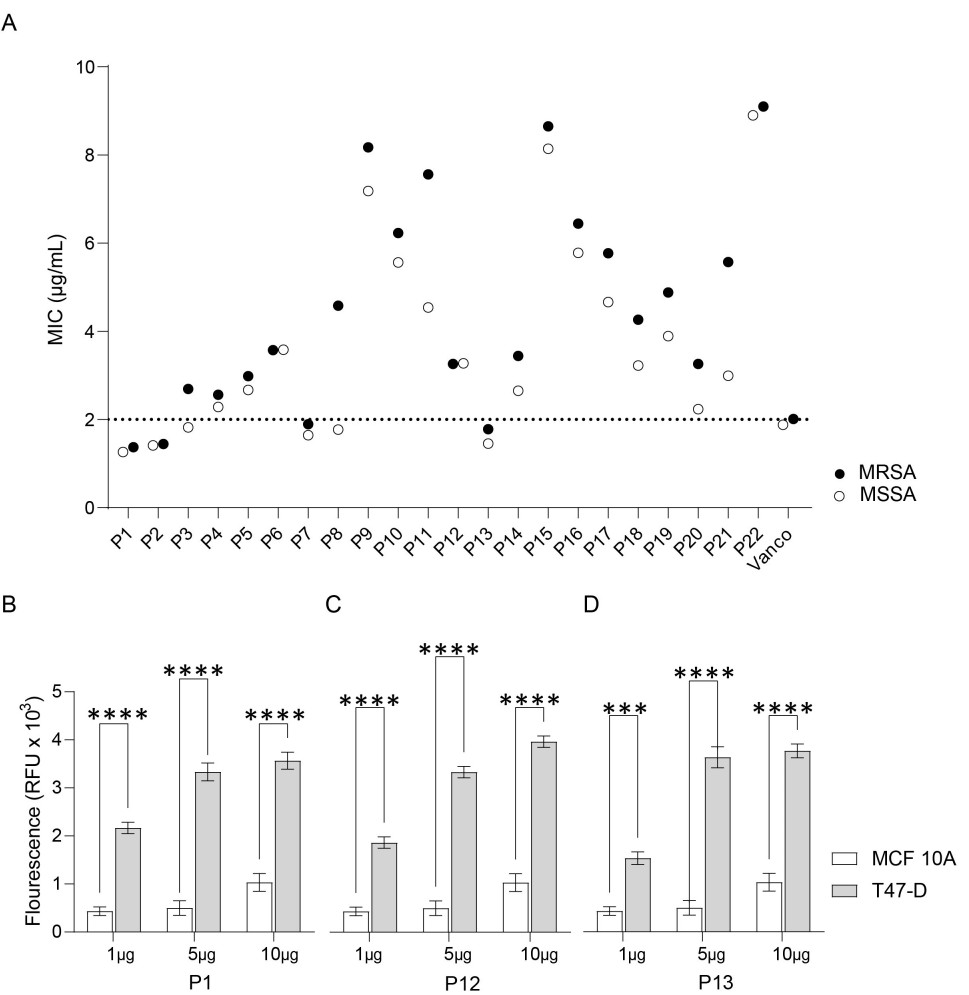

**Fig 2. Antibacterial and antitumor activity of axolotl-derived AMPs.** (A) The minimium inhibitory concentration (MIC) of each peptide measured 24 hours after stimulation against methicillin-resistant *Staphylococcus aureus* (MRSA; black) and methicillin-sensitive *Staphylococcus aureus* (MSSA; white). The MIC of vancomycin, 2 μg/mL, was used as a threshold for identifying inhibitory effects (dashed line). (B-D) The antitumor activity as measured by apoptosis assay after 24-hour stimulation with peptides 1, 12, and 13 in increasing doses (1, 5, and 10 g) against MCF10A (normal breast epithelium) and T47-D (mammary carcinoma) cell lines. RFU; Relative fluorescent units. * p ≤ 0.05, **p ≤ 0.01, ***p ≤ 0.001, ****p ≤ 0.0001.

## Apoptosis assay

Some AMPs have been reported to have anticarcinogenic effects. To test if the AMPs synthesized in our study had these properties, we performed a dose-dependent apoptosis assay following stimulation of T-47D mammary carcinoma cells. Apoptosis was measured following 24 hours of stimulation. The induction of apoptosis varied among the 22 peptides tested, with peptides 1, 12, and 13 showing the most pronounced effects (Fig 2B–D). While a concentration-dependent trend was observed, the differences were not uniformly significant across all concentrations and peptides. At 10 μg/ml, these three peptides exhibited the highest levels of caspase-3/7 activity, indicating stronger apoptotic induction. A similar trend, though less pronounced, was observed at 5 μg/ml. At the lowest concentration of 1 μg/ml, peptides 1 and 12 still showed increased apoptotic activity compared to the control, while peptide 13's effect was less distinct at this concentration.

### Gene expression analysis

Based on the apoptotic effects observed after stimulation with peptides 1, 12 and 13, we wanted to characterize the expressional changes occurring in the mammary carcinoma cells. Gene expression analysis was performed after 24 hours of peptide stimulation. Following stimulation with Peptide 1 we found a significant decrease in the expression of CCND2, CTSD, and IL6, and no upregulation of the gene panel was observed (Fig 3A). Following stimulation with Peptide 12, there was a significant increase in BRCA1, JUN, NR3C1, RB1, SFN, and XBP1, and significant decrease in CCND2, EGF, GLI1, KRT5, MMP2, and SERPINE1 (Fig 3B). Stimulation with Peptide 13 resulted in a significant increase in expression of BRCA1, BRCA2, NR3C1, SERPINE1, and XBP1, and a significant decrease in expression of IL6 and MMP2 (Fig 3C). Overall, it is noticeable that IL6 was downregulated in all cells stimulated with the peptides. CCND2 was downregulated in cells stimulated with peptide 1 and peptide 12. In contrast, MMP2 was downregulated in cells stimulated with peptides 12 and 13. The cells stimulated with peptide 12 or with peptide 13 had in common that BRCA2, NR3C1, SFN, and XBP1 were upregulated in them.

## Discussion

Antibiotic resistance is predicted to rise in the future, making it crucial to discover alternative solutions [37]. Antimicrobial peptides (AMPs) are promising candidates due to their low risk of developing resistance and broad-spectrum activity against bacteria [7]. We hypothesized that the AMPs produced by the axolotl's innate immune system possess antibacterial properties. Additionally, despite high levels of cell division, axolotls exhibit a reduced risk of cancer [38]. Therefore, we hypothesized that skin mucus-produced antimicrobial peptides (AMPs) exhibit antitumor activity.

The skin mucus-derived AMPs were initially characterized, and 22 candidate peptides were identified and ranked based on their likelihood of having antibacterial activities (Table 1, Supl. 2). Synthetic analogs of these peptides were tested against MSSA and MRSA bacteria; six peptides (1, 2, 13, 7, 8, and 3) showed significant growth inhibition towards MSSA and four peptides (1, 2, 13, and 7) against MRSA (Fig 2A, Supl.1). Our investigation yielded several antimicrobial peptides (AMPs) exhibiting significant efficacy against resistant bacterial strains. Notably, peptides 1 and 13 demonstrated remarkably low Minimum Inhibitory Concentration (MIC) values. In comparison to previously characterized AMPs, such as LI14 (MIC range: 2-16 µg/mL against methicillin-resistant *Staphylococcus aureus*), our peptides exhibited comparable or superior antimicrobial activity [39]. Specifically, peptide 1 displayed a MIC of 2 µg/mL against methicillin-resistant *Staphylococcus aureus* (MRSA), which is lower than the MIC values reported for vancomycin against similar MRSA strains, typically ranging from 0.5 to 2 µg/mL. The membrane-disrupting mechanism of these AMPs, distinct from traditional antibiotics, may contribute to their efficacy against resistant strains and potentially mitigate the risk of induced resistance [40]. Collectively, these peptides are promising candidates for addressing multidrug resistant bacteria and should be further tested against additional bacterial strains for antimicrobial activities. The results against MRSA are particularly clinically relevant, as this strain of bacteria is resistant to methicillin and other antibiotics. The prevalence of MRSA is expected to increase with the overuse of antibiotics in both the healthcare and agricultural industries [41].

In addition to showing useful antimicrobial properties, AMPs have also been reported to selectively target tumor cells [24]. In particular, the peptide magainin 2 and its effect in breast cancer and other cancer cell lines has been extensively studied [41]. The mechanisms driving antitumor activity are not fully understood due to the diversity of AMP structures, however it

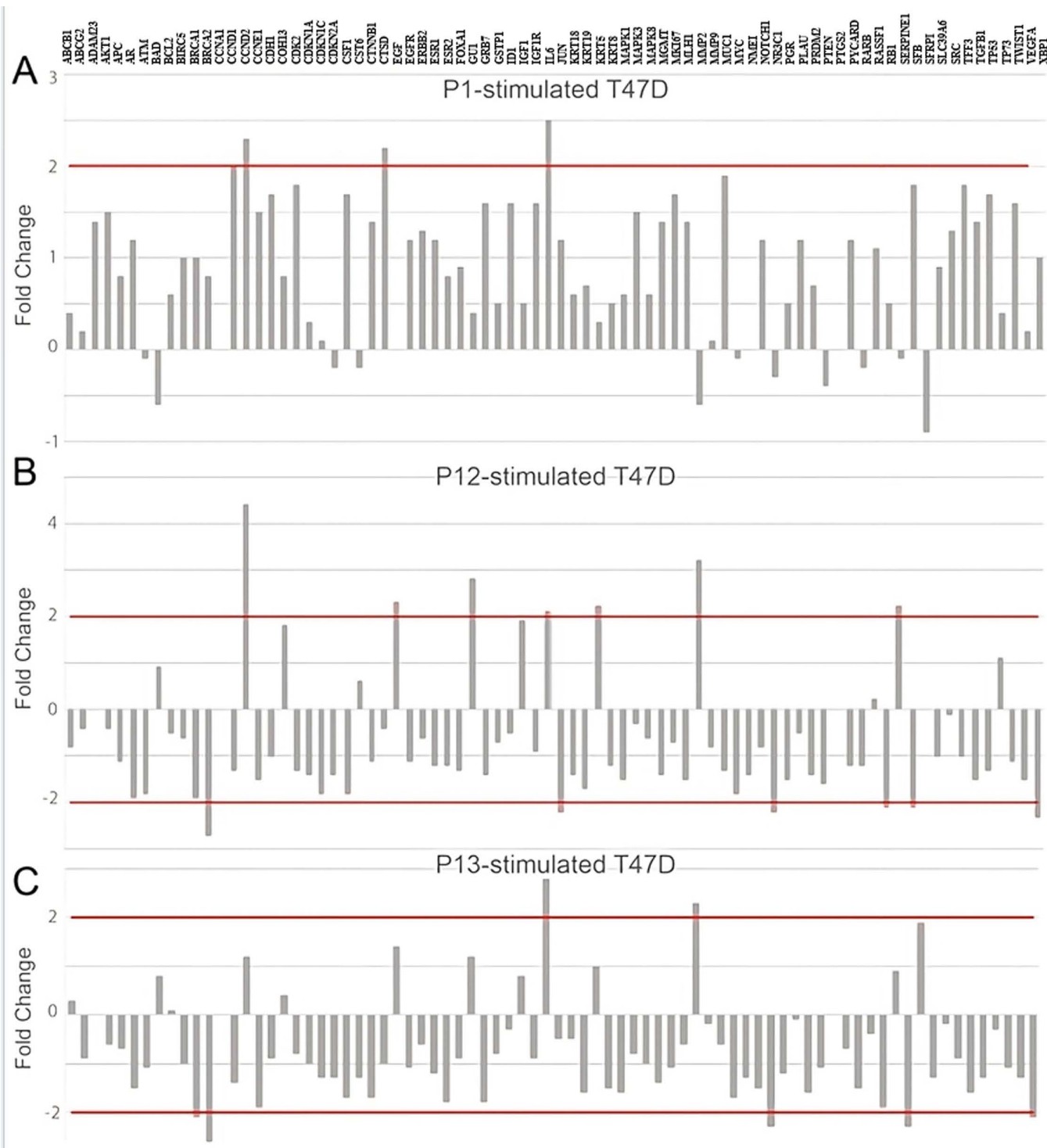

**Fig 3. Gene expression analysis of antitumor activity.** Gene expression was measured in T47-D (mammary carcinoma) cells after 24 hours stimulation with (A) Peptide 1, (B) Peptide 12, and (C) Peptide 13. All peptide stimulations were performed at a concentration of 10 μg/mL, compared to an unstimulated control. Genes were considered upregulated or downregulated if the fold change was above 2.

is thought to be caused by the increased proportion of negatively charged phosphatidylserine present on cancer cells [42].

The study aimed to determine whether any of the peptides had anticancer properties. The impact of AMPs on the apoptosis of breast cancer cell lines T-47D and MF10A was assessed in this study. Peptides 1, 12, and 13 induced significantly increased apoptosis in the cancerous breast tissue cells at all concentrations tested (1μg, 5μg, and 10μg) (Fig 2B). The peptides were observed to specifically target cancer cells without causing cytotoxicity in healthy breast tissue cells (Fig 2B). This indicates that the peptides possess antitumor activity and could be promising candidates for further investigation.

Additionally, upon analyzing the transcriptional changes that occur in these cells after stimulation, we observed a downregulation of several oncogenes and cancer-associated genes, including IL6, MMP2, and CCND2 (Fig 3A–C). IL6 is a so-called pleiotropic cytokine with both tumor-promoting and tumor-inhibiting effects [43]. Interleukin-6 (IL-6) induces an epithelial-mesenchymal transition (EMT) phenotype in human breast cancer cells [44], and direct application of IL-6 to breast cancer cells increases proliferation in estrogen receptor-positive (ER+) cells [45]. Therefore, high levels of IL-6 correlate with a poor prognosis for breast cancer patients [46]. Thus, peptides 1, 12 and 13 may reduce tumor spread and harm cancer cells by down-regulating IL-6.

According to the literature, MMP-2, -3, and -9 may be involved in breast cancer metastasis to the brain [45]. Immunohistochemistry and Western Blots showed significantly higher protein expressions of MMP-2, -3, and -9 in neoplastic brain tissue compared to normal brain tissue [47,48]. Also, gel zymography showed increased MMP-2, and -3 activity in brain metastases [45]. Treatment with the selective MMP inhibitor PD 166793 reduced the development of breast cancer brain metastasis in animals [49]. Thus, downregulation of MMP2 in cells by the peptides could inhibit the metastasis process.

We observed the upregulation of genes associated with positive anti-tumor activity, including NR3C1, BRCA1, BRCA2 and SFN. The glucocorticoid receptor NR3C1 is frequently downregulated in breast tumors, and there is evidence that it plays a role as a tumor suppressor in ER+ breast cancer [50]. NR3C1 was upregulated in cells stimulated with peptides 12 and 13 (Fig 3C,D). The gene products of BRCA1 and BRCA2 are involved in fundamental cellular processes, such as DNA repair and recombination, cell cycle control, and transcription. Mutations in the genes predispose to breast cancer disease [51]. In breast cancer cells, the expression of SFN is silenced by methylations [52]. Thus, the activation of SFN by our identified peptides indicates that the cells are losing their tumor characteristics. Another widely recognized suppressor gene RB1 and is often missing in basal-like breast carcinomas. Its main function is to slow down cell growth [53]. In cells stimulated with peptide 12, RB1 was upregulated.

The peptides upregulate suppressors NR3C1, BRCA1, BRCA2, SFN, and RB1, which may initiate apoptosis processes and growth regulation in breast cancer cells, potentially preventing tumor progression. Additionally, the downregulation of oncogenes and cancer-associated genes, such as IL6, MMP2, and CCND2, may deprive the tumor cell of its growth advantage. These gene-altering peptides could potentially be candidates for breast cancer treatment.

## Limitation

In future experiments, the membrane-perturbing effects of the peptides could be analyzed using microscopic imaging. Additionally, necrotic cell death could be assessed through an LDH assay or PI staining, as well as by measuring intracellular calcium levels. Additional experiments are required and will be performed to substantiate our hypothesis. Screening was performed at 37°C. It is possible that some peptides work better at lower temperatures. Thus, it would be useful to

perform repeat experiments at a temperature which aligns with the body temperature of axolotls. The animals show robustness to infections up to 20°C. Further assays should therefore be performed in this temperature range. The results would not only provide important information regarding applicability in human medicine, but also explain why axolotls become more susceptible to infectious diseases when kept at excessively high temperatures.

## Conclusion

In conclusion, our study identifies AMPs isolated from the skin mucus of the axolotl to exhibit antimicrobial and antitumor activities. This finding supports that these peptides from amphibians can be useful for guiding molecular target discovery. Notably, four peptides found in our study showed significant inhibition of MRSA, and three demonstrated antitumor activity against the T-47D mammary carcinoma cell line. Overall, our findings suggest that these identified AMPs may be promising candidates in the work to address antibiotic resistance and cancer targeting strategies with further validation and studies.

## Supporting information

**S1 Fig.  HPLC chromatograms demonstrating high purity of synthesized compounds. Data tables provide retention times and area percentages for each peak.**
(TIF)

**S2 Fig.  MRSA growth at 37°C over 24 hours, comparing the effects of various peptides, Genta, Vanco, and LL-37 on bacterial growth (OD630nm).**
(PNG)

**S3 Fig.  The figure displays sequences of three synthetic peptides.** Peptide 1 (VAVLGASGGIGQPLSLLLK), Peptide 12 (ILLLcVGEAGDTVQFAEYIQK), and Peptide 13 (FGANALLGVSLAVcKAGAAEK).
(TIF)

## Author contributions

**Conceptualization:** Nadjib Dastagir, Christina Liebsch, Vesna Bucan, Sarah Strauß.

**Data curation:** Nadjib Dastagir, Christina Liebsch, Sabine Wronski, Andreas Pich, Vesna Bucan, Sarah Strauß.

**Investigation:** Jaqueline Kutz.

**Methodology:** Jaqueline Kutz.

**Project administration:** Nadjib Dastagir.

**Resources:** Sabine Wronski, Andreas Pich, Peter Maria Vogt, Sarah Strauß.

**Supervision:** Christina Liebsch, Vesna Bucan, Sarah Strauß.

**Validation:** Vesna Bucan, Sarah Strauß.

**Visualization:** Nadjib Dastagir.

**Writing – original draft:** Nadjib Dastagir.

**Writing – review & editing:** Doha Obed, Peter Maria Vogt, Vesna Bucan, Sarah Strauß.

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
