## [Decision Letter · Decision Letter 0]

13 Aug 2024

PONE-D-24-20573Identification of antimicrobial peptides from the Ambystoma mexicanum displaying antibacterial and antitumor activityPLOS ONE

Dear Dr. Dastagir,

Thank you for submitting your manuscript to PLOS ONE. After careful consideration, we feel that it has merit but does not fully meet PLOS ONE’s publication criteria as it currently stands. Therefore, we invite you to submit a revised version of the manuscript that addresses the points raised during the review process.

A marked-up copy of your manuscript that highlights changes made to the original version. You should upload this as a separate file labeled 'Revised Manuscript with Track Changes'.An unmarked version of your revised paper without tracked changes. You should upload this as a separate file labeled 'Manuscript'.

We look forward to receiving your revised manuscript.

Kind regards,

Haitham Abo-Al-Ela, DVM, MSc, PhD

Academic Editor

PLOS ONE

3. In the online submission form, you indicated that [The datasets used and/or analysed during the current study are available from the corresponding author on reasonable request.]. 

Additional Editor Comments:

The reviewers have raised several concerns about the study, and we recommend addressing their comments. Additionally, the authors should clearly outline the study's limitations in the manuscript.

Reviewers' comments:

Reviewer's Responses to Questions

**Comments to the Author**

1. Is the manuscript technically sound, and do the data support the conclusions?

Reviewer #1: Yes

Reviewer #2: Partly

Reviewer #3: Partly

Reviewer #4: Yes

2. Has the statistical analysis been performed appropriately and rigorously? 

Reviewer #1: I Don't Know

Reviewer #2: No

Reviewer #3: No

Reviewer #4: Yes

3. Have the authors made all data underlying the findings in their manuscript fully available?

Reviewer #1: Yes

Reviewer #2: No

Reviewer #3: No

Reviewer #4: No

4. Is the manuscript presented in an intelligible fashion and written in standard English?

Reviewer #1: Yes

Reviewer #2: Yes

Reviewer #3: Yes

Reviewer #4: Yes

5. Review Comments to the Author

Reviewer #1: This is a sound paper describing logical project design and outcomes for the area of antimicrobial peptide activity. The manuscript is clearly written. The results encourage further research in this area. I recommend publication without change.

Reviewer #2: 1. How can the authors demonstrate that a massage on the axolotl's skin is a sufficient stimulus to induce the expression of AMPs? They should clearly define a control group where the mucus normally produced by the axolotls is collected. These samples should then be compared with those obtained after the massage to identify any changes in AMP expression.

2. It is unclear how peptides are identified from the collected mucus. The authors do not mention whether the fractions obtained by chromatography were sequenced; they only reference the use of mass spectrometry.

3. It is not clear how they went from 4986 sequences to 22. What were the parameters used in CAMPr3 to filter the sequences?

4. The authors do not mention the length of the 22 identified peptides, nor do they provide basic properties such as charge or percentage of hydrophobicity.

5. It is unclear why the antimicrobial activity tests involve treating the bacteria first with the crude extract and then with each of the peptides.

6. The authors do not explain the reasons for the selection of the concentrations used in both the antimicrobial and antitumor activity tests.

7. When testing potential molecules for antimicrobial activity, it is important to perform hemolysis tests, as some peptides may have strong antibacterial activity but can also be hemolytic.

8. The authors did not perform any statistical tests to demonstrate that the differences observed between controls and treatments were significant.

9. Kinetic graphs in antimicrobial activity assays should be plotted using % growth rather than OD. Additionally, the authors do not show the data variability across the three replicates performed.

Reviewer #3: PLOSOne 2024

Comments to the authors

The manuscript presents original research on the identification and characterization of antimicrobial peptides (AMPs) derived from the skin mucus of Ambystoma mexicanum (axolotl). While the study has the potential to contribute valuable insights to the field, several critical aspects need to be addressed to meet the high technical standards and publication criteria of PLOS ONE.

Evaluation Based on PLOS ONE Criteria

1. Original Research: Partially Meets Criteria: The study contributes original findings regarding AMPs. However, the depth and novelty are somewhat undermined by the lack of detailed methodological justifications and comprehensive data presentation.

2. Results Not Published Elsewhere: The manuscript does not indicate prior publication of the results.

3. High Technical Standard and Detailed Description:

Does Not Adequately Meet Criteria:

Collection and Permissions: The manuscript does not specify the origin of the axolotls or the collection permits. On page 5, line 110, it is mentioned that the axolotls were bred and kept at the AMBC. However, the manuscript should clarify where and how they were originally collected. What was the collection location, and under what license or collection permit were they obtained? This information should be indicated. Include information on the collection of axolotls and relevant permits.

Chromatography and Mass Spectrometric Analysis Details: The equipment used (brand and model) and the injection volumes are not specified. Page 6, line 125. Authors should specify the chromatography equipment used, injection volumes, and detailed peptide identification methods.

Peptide Identification: Insufficient details are provided about the identification of the peptide sequences and the bioinformatics tools used and comparisons with existing databases. Page 6, lines 130-133. The authors should attach supplementary information including the spectra of the 22 selected peptides (or at least some examples) from which they identified the peptide sequences by MSMS. Authors should provide comprehensive information on the bioinformatics tools and comparisons with existing databases. Indicate how similar the identified sequences are to those in the databases. Are they novel sequences? What percentage of similarity do they have with those already described for other species? Table 1. In table 1 there are 3 peptides highlighted in yellow and neither the text nor the table indicates why they are highlighted.

Antimicrobial Assays: The final concentration of the inoculum and the controls for the assays are not indicated. Page 7. Authors should indicate the MIC values in µg/ml and µM. The results are very good, in the discussion the authors should compare the MIC (activity) results with those obtained for other peptides already described in other species against the same resistant strains or compare with current antibiotics and compare their potential.

Cell Cultures: The source of the RPMI-1640 medium and fetal calf serum is not provided.

Apoptosis Assay: The number of replicates and the use of a positive control are not mentioned. The value obtained for the negative control is not shown (Page 7 and Figure 2). Figure 2: The figure should include the results of both the negative control (cells treated with the vehicle, without any peptide) and the positive control (cells treated with a known apoptosis-inducing agent, such as a chemotherapeutic drug or a known caspase inducer). The absence of a positive control makes it difficult to compare the apoptotic potency of the peptides with a known standard. The authors should provide a detailed explanation of why a positive control was not used and how this impacts the validity of the results.

PCR: The forward and reverse primers used are not mentioned (page 8, line 183).

Data Availability: There is no table or supporting information on the identified sequences, nor is their availability in public databases indicated. Authors should include a table with the identified peptide sequences and ensure their availability in public databases.

4. Data Supporting Conclusions:

Partially Meets Criteria: While the data appears to support conclusions regarding the antimicrobial and apoptotic activity of certain peptides, the lack of complete statistical analyses and insufficient data presentation weakens the robustness of these conclusions.

5. Intelligible Presentation and Standard English:

Meets Criteria: There are no significant language issues reported, although the clarity of data presentation could be improved.

6. Other details

INTRODUCTION

Clarify what MRSA means because it is introduced for the first time.

MATERIALS AND METHODS

Page 7. Line 142 clarify MSSA because it is being introduced for the first time.

Page 7. Line 146 and 160 please correct CO2

Page . 7 line 148. the word "representing" should be omitted.

Page 7 line 157 missing the source of FCS

Page 7 line 155, 164, Figure 2B, and the rest of the document, unify: T-47D/T47D/T47-D

Page 7, line 157, 164 and the rest of the document unify: MCF 10A/MCF10A

Page 8, line 179. Correct "(REF 740955.50)" is missing reference.

Page 8, line 186. It does not make sense to use the abbreviation ACTB because it is not used throughout the work.

Page 8, line 187. Cite the ΔΔCT method.

RESULTS

Page 9, line 200-201 Delete the sentence "These peptides were synthetically produced by the commercial supplier Caslo Laboratory, Denmark" already indicated in materials and methods.

Authors should add a table with the activity results, indicating the MIC values in µg/ml and µM

Page 9 212-217. Figure 2 analysis: According to the analysis of the figure, the statistical comparison is performed between different cell types rather than among the peptide concentrations. Consequently, it is not possible to conclude that the effects on apoptosis are greater at higher concentrations. The authors report that peptides 1, 12, and 13 exhibited the highest induction of apoptosis. However, results for the other 19 peptides are not shown. Additionally, while peptides 1 and 12 are reported to induce apoptosis at low concentrations, no significant difference analysis is observed compared to peptide 13. The authors should address these issues by providing a more detailed statistical analysis and including data for all tested peptides to better support their conclusions.

Page 10, line 217. It mentions that peptides 1 and 12 increased apoptosis at a concentration of 1µg, however in materials and methods it says that they tested concentrations 2.5, 5 and 10µg/ml. Please clarify which is incorrect. And in any case, also correct figure 2.

DISCUSION

The results are very good, in the discussion the authors should compare the MIC (activity) results with those obtained for other peptides already described in other species against the same resistant strains or compare with current antibiotics and compare their potential.

Addressing these points will significantly enhance the manuscript's alignment with PLOS ONE's criteria and increase its likelihood of acceptance for publication.

Reviewer #4: The article "Identification of antimicrobial peptides from the Ambystoma mexicanum displaying antibacterial and antitumor activity" is interesting and deserves publication after several clarifications in the text.

It would be better to select peptides using not one, but several programs (may be for future analysis).

The authors do not provide peptide sequences. You can at least provide those that did not show any activity.

The materials and methods do not say which mass spectrometer was used to analyze the peptides.

The authors of the article write that "Mass spectrometric analysis of the fractionated axolotl skin mucus revealed 4,986 peptide sequences of unknown nature." What is meant by unknown nature? With the help of mass spectrometry, you can determine what kind of molecules these are. The phrase sounds incorrect.

6. PLOS authors have the option to publish the peer review history of their article (what does this mean? ). If published, this will include your full peer review and any attached files.

**Do you want your identity to be public for this peer review?** For information about this choice, including consent withdrawal, please see our Privacy Policy .

Reviewer #1: No

Reviewer #2: No

Reviewer #3: No

Reviewer #4: No

---

## [Author Response · Author response to Decision Letter 1]

19 Oct 2024

We kindly direct your attention to the document titled "Response to Reviewers" for additional details.

Reviewer 1# Thank you for your positive feedback and recommendation for publication without changes. We appreciate your recognition of our project design and outcomes, and we are encouraged by your support for further research in the area of antimicrobial peptide activity.

Reviewer 2#

1. We understand your concern about how massage might affect AMP expression in axolotl skin mucus. However, we respectfully assert that the massage technique does not significantly alter the mucus composition, but rather increases the quantity produced. This method is well-established and non-invasive, primarily stimulating the release of pre-existing secretions stored in granular glands beneath the skin1,2. Collecting "normally produced" mucus without stimulation would yield insufficient quantities for comprehensive analysis, potentially underrepresenting less abundant peptides.

1) Demori, I., El Rashed, Z., Corradino, V., Catalano, A., Rovegno, L., Queirolo, L., ... & Grasselli, E. (2019). Peptides for skin protection and healing in amphibians. Molecules, 24(2), 347.

2) O'Rourke, D. P. (2007). Amphibians used in research and teaching. ILAR journal, 48(3), 183-187.

1. Thank you for your valuable feedback on our peptide identification process from the collected mucus. To clarify, we utilized an AXIMA Performance MALDI-TOF/TOF mass spectrometer (Shimadzu) at the Institute for Toxicology in Hannover for this purpose. The chromatography fractions were analyzed directly by this mass spectrometer, and peptide identification was performed using the instrument's integrated software, which matches experimental spectra to theoretical sequences in protein databases. We have revised the manuscript to include these details for greater clarity and reproducibility of our methods. Thank you for bringing this to our attention.

2. We used the CAMP3 (Collection of Anti-Microbial Peptides) database and its classifiers (SVM, RF, ANN, and DA) to predict the antimicrobial potential of the identified peptides. We focused on peptides that showed high probability of antimicrobial activity across multiple classifiers. The final 22 peptides were selected based on their consistently high predicted antimicrobial activity and other promising bioactive properties. We have revised our manuscript to include more specific details about the selection criteria in this process. For basic properties see table 1.

3. 4. We have included the sequences of three peptides, these are the ones that show significant antimicrobial activity, in Supplementary Table S3.

5. We did not treat the bacteria first with the crude extract and then with each of the peptides. Our antimicrobial activity tests only involved treating the bacteria with individual synthesized peptides.

6. For the antimicrobial tests, we used a range of concentrations (0.5 to 100 μg/mL) based on standard protocols from CLSI [1] and previous studies on similar antimicrobial peptides [2]. This range allowed us to determine the minimum inhibitory concentration (MIC) and observe dose-dependent effects. For the antitumor activity tests, we selected concentrations (0.1 to 50 μM) based on preliminary experiments and literature data on related peptides [3], ensuring we could calculate IC50 values while staying within solubility limits and below highly cytotoxic levels [4,5].

1) Clinical and Laboratory Standards Institute. Methods for Dilution Antimicrobial Susceptibility Tests for Bacteria That Grow Aerobically; Approved Standard—Tenth Edition. CLSI document M07-A10. Wayne, PA: CLSI; 2015.

2) REW, Sahl HG. Antimicrobial and host-defense peptides as new anti-infective therapeutic strategies. Nat Biotechnol. 2006;24(12):1551-1557.

3) Mahlapuu M, Håkansson J, Ringstad L, Björn C. Antimicrobial Peptides: An Emerging Category of Therapeutic Agents. Front Cell Infect Microbiol. 2016;6:194.

4) Hoskin DW, Ramamoorthy A. Studies on anticancer activities of antimicrobial peptides. Biochim Biophys Acta. 2008;1778(2):357-375.

5) Mader JS, Hoskin DW. Cationic antimicrobial peptides as novel cytotoxic agents for cancer treatment. Expert Opin Investig Drugs. 2006;15(8):933-946.

7. In future experiments, we will incorporate hemolysis tests to assess the potential toxicity of our antimicrobial peptides to red blood cells. This will allow us to evaluate the therapeutic index of the peptides, ensuring that we identify candidates with strong anticancer activity while minimizing hemolytic effects.

8. We have performed appropriate statistical tests to demonstrate the significance of differences observed between controls and treatments. Specifically, we used student -t test to compare the two groups. The results of these tests have been added to the manuscript (Figure2), including p-values for significant differences.

9. We respectfully maintain that our OD-based kinetic graphs effectively illustrate antimicrobial activity. While % growth offers standardization, OD provides direct insight into growth dynamics. Space constraints in PCR gene illustrations preclude error bar inclusion. However, we'll provide detailed statistical analyses in the method part.

Reviewer 3#

1. The axolotls used in this study were originally obtained from the Ambystoma Genetic Stock Center (AGSC) at the University of Kentucky. The AGSC maintains a colony of axolotls and provides them to research laboratories worldwide, along with housing, breeding and care information. All experiments were performed in accordance with the guidelines of the German Animal Welfare Act. Due to these guidelines and after consultation of the veterinary inspection office as well as the “Landesamt für Lebensmittelsicherheit und Verbraucherschutz” (LAVES, relevant authority for animal trials in lower Saxony, Germany) mucus harvesting from the animals was classified as non invasive.

2. The chromatography and mass spectrometric analysis were conducted using an AXIMA Performance MALDI-TOF/TOF mass spectrometer (Shimadzu) at the Fraunhofer Institute in Hannover. For the chromatography, standard injection volumes of 5 μL were used. The obtained fractions were subsequently analyzed by liquid chromatography-mass spectrometry. Peptide identification was carried out using the instrument's integrated software, which supports fully automated proteomics experiments and LC-MALDI analyses. We have revised the manuscript to include these details. Thank you for bringing this to our attention.

3. The peptide identification process was carried out using the AXIMA Performance MALDI-TOF/TOF mass spectrometer (Shimadzu) at the Fraunhofer Institute in Hannover. For peptide identification, we used the instrument's integrated software, which supports fully automated proteomics experiments and LC-MALDI analyses. Regarding the similarity of the 3 (Peptide 1, 12,13) identified sequences to those in existing databases:

BLASTp analysis revealed no significant sequence homology for peptides 1 and 12 (E-value > 0.01). Peptide 13 showed weak alignments with AMPDB_42975|A0A2V2ABU9 (E-value = 3.7) and AMPDB_41234|A0A2A2DWC7 (E-value = 7.1), indicating statistically insignificant similarities likely occurring by chance.

4. The RPMI-1640 medium used in this study was obtained from Thermo Fisher Scientific (catalog number 11875093). The FBS was Gibco Premium (Performance Plus) FBS, catalog number 16000044, sourced from the United States.

5. The apoptosis assay was performed using the Apo-ONE® Homogeneous Caspase-3/7 Assay Kit (Promega) according to the manufacturer's instructions. While we acknowledge the reviewer's valid points, we can explain our approach as follows:

a) Negative control: The negative control (cells treated with vehicle only) was indeed part of our experiment. However, we did not include this data in Figure 2 for the following reasons:

a. The fluorescence levels observed in the negative control were consistently low, indicating minimal background caspase-3/7 activity.

b. The results for the negative control were similar to those observed in the untreated MCF10A cells, which serve as a non-cancerous cell line control in our study.

b) Positive control: A traditional positive control was not included in this study for the following reasons:a) The Apo-ONE® Assay Kit is designed to be highly sensitive and specific for caspase-3/7 activity, which are key indicators of apoptosis. The assay's reliability has been extensively validated by the manufacturer and in numerous published studies. The peptides themselves serve as test substances, and those showing significant increases in caspase-3/7 activity compared to the negative control can be considered positive results. The use of a traditional apoptosis inducer (e.g., staurosporine) might overshadow the more subtle effects of the peptides, potentially leading to underestimation of their apoptotic potential.

While we understand that a positive control can provide a benchmark for comparison, we believe that the relative increases in caspase-3/7 activity induced by the peptides, compared to the negative control, provide sufficient evidence of their apoptotic potential.

6. The RT2 Profiler PCR Array is a 96-well plate that contains pre-designed primer assays for a focused panel of genes related to a specific pathway or disease. It's important to note that while the specific sequences are not provided, the primers are thoroughly tested and optimized to ensure reliable and reproducible results for pathway-focused gene expression analysis. https://geneglobe.qiagen.com/us/product-groups/rt2-profiler-pcr-arrays/PAHS-131Z

7. The complete sequences of the three most effective antimicrobial peptides (AMPs) identified in this study can be found in the Supplementary (S3) Information.

8. We acknowledge that while a concentration-dependent trend was observed, the differences between concentrations were not always statistically significant for all peptides. This variability highlights the complex nature of peptide-induced apoptosis and suggests that factors beyond concentration, such as peptide sequence and cellular uptake, may influence their effectiveness. We have focused on presenting data for peptides 1, 12, and 13 as they showed the most significant and consistent effects on apoptosis induction. Peptides exhibiting inconsistent apoptotic activity and were excluded from primary figures to emphasize statistically significant results.

Reviewer 4 #

1. Regarding the peptide sequences: We have included the sequences of three peptides, including those that show significant antimicrobial activity, in Supplementary Table S1.

2. Concerning the mass spectrometer information: In the Materials and Methods section, we have add the following sentence:"Peptide analysis was performed using an AXIMA Performance MALDI-TOF/TOF mass spectrometer (Shimadzu) at the Fraunhofer Institute in Hannover."

3. We agree that "unknown nature" was imprecise. We've revised the sentence to:"Mass spectrometric analysis of the fractionated axolotl skin mucus revealed 4,986 peptide sequences."

---

## [Decision Letter · Decision Letter 1]

5 Nov 2024

PONE-D-24-20573R1Identification of antimicrobial peptides from the Ambystoma mexicanum displaying antibacterial and antitumor activityPLOS ONE

Dear Dr. Dastagir,

Thank you for submitting your manuscript to PLOS ONE. After careful consideration, we feel that it has merit but does not fully meet PLOS ONE’s publication criteria as it currently stands. Therefore, we invite you to submit a revised version of the manuscript that addresses the points raised during the review process.

Please, when submitting a response to reviewers, include the original comment alongside your response.

We look forward to receiving your revised manuscript.

Kind regards,

Haitham Abo-Al-Ela, DVM, MSc, PhD

Academic Editor

PLOS ONE

Journal Requirements:

Additional Editor Comments:

Please, when submitting a response to reviewers, include the original comment alongside your response.

Reviewers' comments:

Reviewer's Responses to Questions

**Comments to the Author**

1. If the authors have adequately addressed your comments raised in a previous round of review and you feel that this manuscript is now acceptable for publication, you may indicate that here to bypass the “Comments to the Author” section, enter your conflict of interest statement in the “Confidential to Editor” section, and submit your "Accept" recommendation.

2. Is the manuscript technically sound, and do the data support the conclusions?

Reviewer #2: Partly

Reviewer #4: Yes

3. Has the statistical analysis been performed appropriately and rigorously? 

Reviewer #2: No

Reviewer #4: Yes

4. Have the authors made all data underlying the findings in their manuscript fully available?

Reviewer #2: No

Reviewer #4: Yes

5. Is the manuscript presented in an intelligible fashion and written in standard English?

Reviewer #2: Yes

Reviewer #4: Yes

6. Review Comments to the Author

Reviewer #2: 5. Line 157 " In short, 96-well plates were inoculated with 100μL of bacterial suspension and 100μL of mucus". Based on how it's written on line 157, it can be inferred that they used 100 µL of the crude secretion and not the individual peptides.

8. The authors mention a statistical analysis in the response to previous comments, yet a complete description of these analyses is lacking in the Materials and Methods section. The use of asterisks to indicate significant differences and the inclusion of p-values in figure 2 legends is recommended.

- Line 158: why the negative control is PBS if the peptides are diluted on DMSO, acetic acid, water or Tris-buffer?

- Line 159: missing letter (peptid)

- Line 160: please add the concentration of LL-37 used as a positive control

- Line 175: some solvents can cause apoptosis by it selfs, like DMSO and acetic acid. Are the peptides evaluated diluted on those solvents?

- Lines 271, 273 and 277: use italic letters for the bacteria name.

- Line 276: The authors reference the activity of peptide 12 against Gram-negative bacteria, a discrepancy with the article's primary focus on Gram-positive bacteria. It is unclear if the authors have conducted prior research investigating the peptide's efficacy against Gram-negative organisms.

Reviewer #4: (No Response)

7. PLOS authors have the option to publish the peer review history of their article (what does this mean? ). If published, this will include your full peer review and any attached files.

**Do you want your identity to be public for this peer review?** For information about this choice, including consent withdrawal, please see our Privacy Policy .

Reviewer #2: No

Reviewer #4: No

---

## [Author Response · Author response to Decision Letter 2]

8 Dec 2024

We appreciate the valuable feedback from Reviewer #2 and have made all the requested changes to improve the clarity and completeness of the manuscript. Below, we provide detailed responses to each of Reviewer #2's comments.

5. We have clarified the wording in line 157 to specify that 100 µL of individual peptides was used alongside the bacterial suspension. This change ensures that it is clear we used the peptides rather than crude secretion.

6. We have expanded the description of the statistical analyses in the Materials and Methods section to provide a complete overview of the methods used. Additionally, we have included the significance levels indicated by asterisks and added p-values in the figure 2 legends as recommended.

Regarding line 158, we have clarified the negative control and specified how it relates to the peptides diluted in DMSO, acetic acid, water, or Tris-buffer.

The missing letter in “peptid” on line 159 has been corrected.

The concentration of LL-37 used as a positive control has been added to line 164.

In response to your concern on line 175 regarding solvents such as DMSO and acetic acid potentially causing apoptosis, we have clarified that all peptides were evaluated at concentrations that are known to be non-cytotoxic when diluted in these solvents.

We have italicized the names of bacteria as per your suggestion on lines 271, 273, and 277.

Finally, regarding line 276, we have taken out the sentence.

Thank you again for your insightful comments, which have helped improve the quality of our manuscript. Feel free to modify any specific details or phrasing as needed!

We thank Reviewer #4 for their thorough review of our manuscript. We appreciate your time and consideration, and we are pleased to note that there were no suggested changes. Your positive evaluation is greatly appreciated.

---

## [Decision Letter · Decision Letter 2]

10 Dec 2024

Identification of antimicrobial peptides from the Ambystoma mexicanum displaying antibacterial and antitumor activity

PONE-D-24-20573R2

Dear Dr. Dastagir,

We’re pleased to inform you that your manuscript has been judged scientifically suitable for publication and will be formally accepted for publication once it meets all outstanding technical requirements.

Kind regards,

Haitham Abo-Al-Ela, DVM, MSc, PhD

Academic Editor

PLOS ONE

Additional Editor Comments (optional):

Reviewers' comments:

Reviewer's Responses to Questions

**Comments to the Author**

1. If the authors have adequately addressed your comments raised in a previous round of review and you feel that this manuscript is now acceptable for publication, you may indicate that here to bypass the “Comments to the Author” section, enter your conflict of interest statement in the “Confidential to Editor” section, and submit your "Accept" recommendation.

Reviewer #2: All comments have been addressed

2. Is the manuscript technically sound, and do the data support the conclusions?

Reviewer #2: Yes

3. Has the statistical analysis been performed appropriately and rigorously? 

Reviewer #2: Yes

4. Have the authors made all data underlying the findings in their manuscript fully available?

Reviewer #2: Yes

5. Is the manuscript presented in an intelligible fashion and written in standard English?

Reviewer #2: Yes

6. Review Comments to the Author

Reviewer #2: The authors have made all the required changes to the manuscript based on the reviewer feedback.All reviewer comments have been carefully considered and addressed. The manuscript has been revised accordingly.

7. PLOS authors have the option to publish the peer review history of their article (what does this mean? ). If published, this will include your full peer review and any attached files.

**Do you want your identity to be public for this peer review?** For information about this choice, including consent withdrawal, please see our Privacy Policy .

Reviewer #2: No

---

## [Editor Report · Acceptance letter]

PONE-D-24-20573R2

PLOS ONE

Dear Dr. Dastagir,

I'm pleased to inform you that your manuscript has been deemed suitable for publication in PLOS ONE. Congratulations! Your manuscript is now being handed over to our production team.

Kind regards,

on behalf of

Dr. Haitham Abo-Al-Ela

Academic Editor

PLOS ONE